# Nursing Interventions to Prevent Secondary Injury in Critically Ill Patients with Traumatic Brain Injury: A Scoping Review

**DOI:** 10.3390/jcm13082396

**Published:** 2024-04-19

**Authors:** Rita Figueiredo, Cidália Castro, Júlio Belo Fernandes

**Affiliations:** 1Department of Nursing, Almada-Seixal Local Health Unit, 2805-267 Almada, Portugal; figueiredor1701@gmail.com; 2Nurs * Lab, Caparica, 2829-511 Almada, Portugal; jfernandes@egasmoniz.edu.pt; 3Egas Moniz Center for Interdisciplinary Research (CiiEM), Egas Moniz School of Health & Science, Caparica, 2829-511 Almada, Portugal

**Keywords:** secondary injury prevention, traumatic brain injury, nursing, critically ill

## Abstract

**Background:** Traumatic brain injury is a prevalent health issue with significant social and economic impacts. Nursing interventions are crucial in preventing secondary injury and improving patient prognosis. This scoping seeks to map and analyze the existing scientific evidence on nursing interventions aimed at preventing secondary injuries in critically ill patients with traumatic brain injury. **Methods:** The review was conducted according to Arksey and O’Malley’s methodological framework. The electronic databases Pubmed, MEDLINE Complete, CINAHL Complete, Nursing & Allied Health Collection: Comprehensive, Cochrane Central Register of Controlled Trials, and Cochrane Clinical Answers were consulted in May 2023. We included articles published in English and Portuguese between 2010 and 2023. **Results:** From the initial search, 277 articles were identified, with 15 meeting the inclusion criteria for the review. Nursing interventions for TBI patients include neuromonitoring, therapeutics, analytical surveillance, professional training, and family support. Nurses play a crucial role in detecting neurological changes, administering treatments, monitoring metabolic markers, training staff, and involving families. These interventions aim to prevent secondary injury and improve patient outcomes. **Conclusions:** By prioritizing evidence-based practice and utilizing innovative technologies, nurses enhance TBI patient care and contribute to overall well-being.

## 1. Introduction

Traumatic brain injury (TBI) is characterized as a disruption in cerebral function, whether transient or enduring, stemming from an external force that induces swift movement of the brain within the skull [1].

Primary injury occurs at the moment of impact and is irreversible, initiating a sequence of cellular processes recognized as secondary injury [2]. While the initial trauma inflicts damage, the succession of secondary events significantly influences the outcome for critically ill patients [1].

Several authors advocate for a window of opportunity to enhance outcomes in critically ill patients’ victims of TBI through early therapeutic interventions, as the mechanisms underlying secondary injury only become irreversible after a specific timeframe [3].

Secondary injury begins with inflammatory processes, subsequent blood–brain barrier disruption, cerebral edema, and the liberation of inflammatory mediators [4]. The primary culprits behind secondary injury encompass cerebral edema, elevated intracranial pressure, hypoperfusion, and hypoxemia, culminating in cerebral tissue compression and subsequent damage [5]. Additionally, electrolyte imbalances, notably altered sodium levels, and exacerbate neurological deficits by precipitating significant shifts in water balance [6].

TBI is frequently labeled as a silent epidemic, emerging as a leading cause of both mortality and morbidity among young individuals across both developed and developing nations, driven partly by the rising prevalence of motorcycle usage [7].

The World Health Organization et al. [8] classify trauma as a significant public health issue, noting that each year, 5.8 million people succumb to trauma, leaving many debilitated. Expanding on this, Krenz et al. [9] assert that TBI can be compared to a chronic disease due to its enduring consequences, significantly affecting both the individual’s and their family’s quality of life and socioeconomic status. In Portugal, statistics from 2022 revealed approximately 32,788 accidents resulting in 40,699 injuries, with 17,002 attributed to collisions and 11,301 to road accidents [10].

Efforts to mitigate primary injury primarily focus on preventive measures, such as public awareness campaigns promoting helmet use. However, therapeutic interventions are crucial in modulating secondary injury [3]. The imperative to prevent secondary injury is evident, aiming to reduce the onset of comorbidities associated with TBI. The management of TBI patients necessitates a multidisciplinary team with expertise in brain injury pathophysiology and the cascade of secondary injuries. This knowledge forms a critical foundation for therapeutic decisions, especially in areas with limited evidence [11]. The primary focus of therapeutic interventions should be on addressing the primary injury and minimizing neurological damage resulting from secondary injury.

Nursing professionals play a pivotal role within the multidisciplinary team, influencing the outcomes of critically ill patients due to their proximity in identifying neurological deficits and intervening promptly [12]. Indeed, the authors argue that regular neurological assessments and prompt recognition of shifts in clinical status can significantly impact outcomes, potentially tipping the balance between life and death. Delayed diagnosis of complications can precipitate a deterioration in the critically ill patients’ clinical condition, limiting treatment modalities and, ultimately, culminating in irreparable damage [13]. The primary objective of nursing interventions for TBI patients is to prevent and minimize secondary brain injury, which can lead to physical and cognitive complications [14]. Therefore, nurses must have current scientific knowledge, professional expertise, proficiency in performing procedures, and the ability to prioritize and make decisions effectively [15].

In addition to direct patient care, supporting the victim’s family is crucial. During this vulnerable phase, family members often experience distress and uncertainty about the patient’s future, fearing the possibility of losing their loved one [16].

Nursing interventions are crucial in promoting recovery and mitigating secondary injury among critically ill patients with TBI [17]. Continuous training for nursing professionals is imperative to equip them with the requisite skills to reduce risks and enhance the benefits of specialized interventions. Iavagnilio [13] further underscores this notion, contending that prevention and training are pivotal factors influencing outcomes in critically ill patients’ victims of TBI. The overarching goal remains to enhance care quality, achieved through integrating acquired knowledge into practice, with nursing research as the bedrock of evidence-based practice. 

This scoping review aims to map and analyze the existing scientific evidence on nursing interventions aimed at preventing secondary injuries in critically ill patients with TBI.

## 2. Methodology

The current scoping review adhered to the methodological framework outlined by Arksey and O’Malley [18]. The Preferred Reporting Items for Systematic Reviews and Meta-Analyses Extension for Scoping Reviews (PRISMA-ScR) was employed to enhance comprehension, uphold transparency, and safeguard the review’s quality.

### 2.1. Phase 1: Identification of the Research Question

In formulating the research question, the mnemonic “PCC” was used by the recommendations of Joanne Briggs Institute [19] for scoping reviews, translating into Participants, Concept, and Context. This results in the following research question: what nursing interventions prevent secondary injuries in critically ill patients with TBI?

### 2.2. Phase 2: Identification of Relevant Studies

A search was conducted on the EBSCOhost search engine, encompassing the following databases: MEDLINE Complete, CINAHL Complete, Nursing & Allied Health Collection: Comprehensive, Cochrane Central Register of Controlled Trials, and Cochrane Clinical Answers, yielding a total of 156 articles. A parallel inquiry was conducted on PubMed to complement the search, employing identical criteria and identifying 129 additional articles. Thus, a total of 277 articles were amassed for analysis. This comprehensive search was executed on 4 May 2023.

Indexed Medical Subject Headings were utilized alongside Boolean operators to construct the following search string: (nurs *) AND [(critical care) OR (intensive care) OR (Intensive Care Unit)] AND [(head injury), OR (brain injury) OR (TBI)].

The inclusion and exclusion criteria were established following the PCC mnemonic (Participant, Concept, and Context) (Table 1).

The search was limited to articles published between 2010 and 2023. 

Given constraints in translation resources, the review excluded papers published in languages other than English or Portuguese.

### 2.3. Phase 3: Study Selection

The initial search in the selected databases already incorporated filters corresponding to the inclusion criteria. However, some articles were unavailable in full text and consequently rejected, along with duplicate articles identified on the Rayyan Artificial Intelligence-assisted systematic literature review tool.

Subsequently, the articles underwent screening by two independent researchers who assessed the title and abstract. To ensure that only articles pertinent to the research question were screened, all articles were read in full, adhering to the previously defined inclusion criteria. This rigorous process ensured the compilation of articles addressing nursing interventions to prevent secondary injuries in critically ill patients with TBI.

The existing scientific evidence on specialized nursing interventions was mapped and analyzed to prevent the development of secondary injuries in critically ill people who have suffered from TBI.

A flowchart, adapted from PRISMA-ScR, was created (Figure 1) to illustrate the selection of articles at different stages throughout the identification, screening, and selection of studies.

### 2.4. Phase 4: Data Recording

Committed to this review’s objective and research question, two researchers developed an instrument to facilitate a logical and summarized resume of the relevant data described in Table 1.

The extracted data encompassed general information (article title, authors, publication year, and country), methodological details (study design and objective), and findings (nursing interventions).

### 2.5. Phase 5: Collection, Summary, and Presentation of Data

In organizing and summarizing the data, we employed thematic analysis guided by the framework established by Braun et al. [20].

Two researchers conducted independent data analysis. The results are presented using descriptive text supplemented by tables that underscore the significance of nursing interventions. These tables offer a structured overview of key findings, enhancing clarity and facilitating understanding of the role of nursing interventions.

## 3. Results

The research conducted for this scoping review initially identified 277 articles. Following the removal of duplicate documents, 216 unique titles underwent analysis. Subsequently, 26 of these titles were chosen for full-text reading. Finally, 15 articles met the eligibility criteria after completing the article selection process. 

Most selected articles originated from the United States of America, accounting for 66.7% of the total [6,12,13,17,21,22,23,24,25,26]. The remaining articles were from the Netherlands [27], the Republic of Korea [28], Iran [29], and Egypt [7,30], Regarding the methodology of the identified studies, the following breakdown was observed: five literature reviews [6,12,21,26,27], three case studies [13,24,25], one qualitative study [22], three quasi-experimental studies [7,29,30], two quantitative studies [17,28], and one guideline [23].

Table 2 summarizes the included articles, consolidating their most significant characteristics and results.

Data analysis revealed various nursing interventions in the prevention of secondary injury in critically ill patients who are victims of TBI. The interventions were grouped into five categories:

### 3.1. Neuromonitoring

Neuromonitoring involves continuously monitoring critically ill patients’ neurological functions to assess brain activity and promptly detect any changes that may impact the nervous system. This proactive approach enables early adjustments in interventions, ensuring the safety of critically ill patients. Neuromonitoring is crucial for preventing secondary injuries resulting from ischemia and hypoxia [21]. Initially utilized solely for problem detection, neuromonitoring now identifies changes before significant issues arise, facilitating immediate interventions and preventing secondary injuries [21].

Most complications result from increased intracranial pressure [12]. Therefore, nurses must monitor intracranial pressure, which is considered one of the most critical physiological parameters [23]. Authors concur with Cecil et al. [21], considering increased intracranial pressure monitoring and calculating cerebral perfusion pressure as the rule of thumb. They additionally advocate for monitoring cerebral blood flow and oxygenation of brain tissue, mainly using Licox PbtO2 (partial pressure of oxygen), as well as monitoring brain temperature and hypothermia. PbtO2 values are a marker of cerebral ischemia and secondary injury, indicating hypoxia events quicker than intracranial pressure and cerebral perfusion pressure assessment, leading to better outcomes [21]. The ideal PbtO2 reference value is greater than 20 mmHg, with 30 mmHg being the ideal. Methods to improve these values in ischemia conditions include maximizing cerebral blood flow by reducing intracranial pressure through barbiturate administration, cerebrospinal fluid drainage, and craniotomy.

Increased intracranial pressure affects the brain and nervous system and impacts other organs [12], leading to morbidity and mortality in the Intensive Care Unit [23]. Nurses play a crucial role in identifying the causes of increased intracranial pressure and initiating timely treatment, which is essential for preventing injury exacerbations [23].

An increase in intracranial pressure is directly related to the decline in consciousness of critically ill patients, affecting their ability to protect the airway, oxygenate, or ventilate effectively, often necessitating invasive mechanical ventilation [12]. Orotracheal intubation is recommended for critically ill patients with TBI. Non-invasive mechanical ventilation is often ineffective and not recommended, as airway manipulation can irritate vomiting and cough reflexes, leading to increased PaCO_2_ and subsequently increased blood pressure, putting patients at risk of complications due to rising intracranial pressure and bleeding [12]. However, routine nursing interventions such as hygiene care and bronchial secretion suctioning, as evidenced by Nyholm et al. [17], may have the opposite effect, leading to significant increases in intracranial pressure and potential secondary injury.

Nurses can identify deterioration, especially in critically ill patients with reduced consciousness levels, by monitoring pupillary changes. Utilizing a pupillometer is a quick and straightforward method, serving as a valuable tool in assessing neurological status in such patients with TBI [21]. Notably, a constriction velocity of less than 0.8 mm/second indicates increased brain volume, while less than 0.6 mm/second suggests elevated intracranial pressure. Less than 10% pupillary reactivity following light stimulus suggests increased intracranial pressure.

Vasospasm, characterized by cerebral artery constriction, can reduce cerebral blood flow and cause cerebral ischemia. It is challenging to detect, and transcranial Doppler is the method of choice. It allows blood velocity assessment in the middle cerebral artery and internal carotid artery, and it is used routinely in some units [12]. Transcranial Doppler sonography is also used to detect hemodynamic changes in the brain after trauma, including hyperemia, reduced cerebral blood flow, and intracranial hypertension [21].

Seiler et al. [24] recommend continuous electroencephalography (EEG) for early vasospasm detection. EEG aids nurses in detecting subclinical seizures, leading to early intervention. According to the authors, one-third of neurocritical Intensive Care Unit patients experience subclinical seizures, increasing the risk of adverse outcomes.

Fever is a primary complication of aneurysmal subarachnoid hemorrhage, often related to cytokine release in the first 72 h [12]. Temperature monitoring is included in many catheters used for intracranial pressure assessment. Monitoring brain temperature and maintaining hypothermia are neuroprotective nursing interventions that aim to lower brain metabolic demand, which may mitigate secondary neuronal injury and enhance patient outcomes [21].

### 3.2. Therapeutics

Several key elements are essential for controlling intracranial pressure, including fluid therapy, blood pressure regulation, antiepileptic medication, positioning, and sedation [23]. Boling and Groves [12] emphasize that changes in intravascular volume can exacerbate patient status, making regular volume monitoring indispensable. Nurses play a pivotal role in preparing, administering, and monitoring these therapeutic interventions. They ensure accurate preparation and administration of medications, closely monitor patient responses, and promptly address any adverse effects or complications. Additionally, nurses provide ongoing assessment and documentation of therapeutic outcomes, collaborating with the multidisciplinary team to optimize patient care and outcomes. Their vigilance and expertise in therapeutic interventions significantly contribute to managing intracranial pressure and overall patient well-being in the clinical setting.

In terms of fluid therapy, normovolemia is the ideal goal, with isotonic crystalloids recommended over hypotonic fluids in large volumes [12,23,27]. Hydrocortisone and fludrocortisone may be administered to critically ill patients requiring high-volume replacement [12].

Although crystalloids are preferred, hyperosmolar agents such as hypertonic saline or mannitol may be necessary in boluses. Tran [26] highlights that hypertonic saline remains a neuroprotective measure due to its osmotic capacity, which mobilizes fluids from intracellular to intravascular spaces, reducing cerebral edema. Moreover, hypertonic saline helps normalize blood pressure values and increases cardiac output by expanding intravascular volume, resulting in increased mean arterial pressure (MAP) and decreased intracranial pressure, leading to improved cerebral perfusion pressure and cerebral blood flow. Maintaining an ideal cerebral perfusion pressure, typically above 60 mmHg, is crucial and may require vasopressors [23]. However, the lack of consensus among experts on the best therapeutic option hinders the standardization of procedures and the development of action protocols [23].

Oral nimodipine has been associated with improved outcomes in individuals with subarachnoid hemorrhage, although the precise mechanism remains unclear. It limits late neurological deterioration [12].

Statins act as anti-inflammatory agents, reducing the production of inflammatory cytokines and oxidative stress and regulating microglial activation, thereby preserving blood–brain membrane integrity [26].

Sedation and analgesia are recommended for prophylaxis against increased intracranial pressure, particularly with a baseline value above 20 mmHg [17].

However, the efficacy of antipyretics is limited, as they act on the hypothalamus. Their effectiveness depends on intact thermoregulation, which may be compromised in brain injury victims. Cooling blankets can be used, although they often induce shivering, increase metabolic demand, reduce cerebral oxygenation, and necessitate continuous sedation [12].

### 3.3. Analytical Surveillance

Cerebral microdialysis technology is an emerging method for assessing metabolic markers like glucose, pyruvate, lactate, and glycerol. This monitoring entails placing a semipermeable catheter in the brain parenchyma via craniotomy [21]. This technology can aid nurses in assessing metabolic markers in real time, enabling prompt interventions to maintain optimal metabolic balance and prevent further neurological compromise. Maintaining blood glucose levels between 110 mg/dL and 180 mg/dL is crucial to prevent systemic hypoglycemia, metabolic crises, and secondary injury [21]. The authors also highlight the early initiation of nutritional support.

Recent recommendations suggest a more comprehensive glucose maintenance range of 80–200 mg/dL, mitigating the risk of hypoglycemia [12]. Oh et al. [28] observed that TBI patients with poorer recovery had higher blood glucose levels, especially within the initial 72 h, with persistent and fluctuating hyperglycemia, which Boling and Groves [12] associate with vasospasm. However, consensus on the ideal blood glucose levels remains elusive.

Dysnatremias are common in acute brain injury, often due to the central nervous system’s role in regulating sodium and water balance. Hyponatremia, a prevalent electrolyte disorder after TBI, can result from a syndrome of inappropriate secretion of antidiuretic hormone, cerebral salt-wasting syndrome, or iatrogenic causes such as fluid administration or the use of mannitol, corticosteroids, or diuretics to manage cerebral edema [12].

Inappropriate secretion of antidiuretic hormone involves continuous secretion of high antidiuretic hormone levels, leading to water reabsorption, concentrated urine, and hyponatremia. Diagnosis includes low sodium levels, serum osmolarity, and urinary osmolality. Treatment involves fluid restriction and slow correction of sodium levels using hypertonic saline [6].

Cerebral salt-wasting syndrome results from sodium loss and intravascular fluid depletion [6]. While its pathophysiology is not fully elucidated, laboratory values manifest as hyponatremia with low serum osmolarity and high urinary osmolarity. Given their opposing treatments, distinguishing between these syndromes is critical. Inappropriate secretion of antidiuretic hormone necessitates fluid restriction, whereas cerebral salt-wasting syndrome requires volume replacement with saline solution and 3% hypertonic sodium chloride [6].

Hypernatremia is linked to central diabetes insipidus or mannitol use. Central diabetes insipidus involves decreased antidiuretic hormone secretion, leading to severe dehydration and exacerbated electrolyte imbalance if not promptly corrected [6]. Diagnosis relies on signs, symptoms, and analytical values, including polyuria, low urinary osmolarity, high serum osmolarity, and hypernatremia. Correction involves administering exogenous antidiuretic hormone, commonly desmopressin and vasopressin, increasing administered volume if feasible, encouraging water intake via the enteric route, or elevating hypotonic fluid administration.

### 3.4. Professional Training

Intensive care nurses play a crucial role in promptly recognizing neurological decline and providing rapid intervention for TBI victims, reducing the likelihood of secondary injury development [12]. Thus, nurses must prioritize evidence-based practice in their approach.

Nurses possess the potential to positively or negatively influence the risk of secondary injury development based on their scientific knowledge when handling TBI cases. Various interventions, such as suctioning bronchial secretions and changes in positioning, can impact physiological variables, increasing blood pressure and intracranial pressure while decreasing oxygenation and cerebral perfusion pressure [22]. Nurses are adept at recognizing physiological value fluctuations and responding accordingly, perceiving oxygen levels, intracranial pressure, and cerebral perfusion pressure as critical factors in secondary injury development, enabling autonomous interventions [22].

Unfortunately, screening for hyperthermia can be neglected despite its significant contribution to secondary injury development [22]. Delayed diagnosis can exacerbate conditions, limiting treatment options and leading to comorbidities [13]. Therefore, there is an urgent need to develop evidence-based guidelines for healthcare professionals involved in TBI patient care [13].

Mohammed et al. [30] underscored the benefits of implementing clinical pathways in reducing hospitalization-related complications, such as intensive care unit stay and readmission rates, and enhancing patient satisfaction when executed by knowledgeable professionals.

Nevertheless, even with protocol-based guidelines in place, nurses may lack awareness of them, as evidenced by Mcnett et al. [22]. Their study revealed a knowledge gap among nursing teams regarding Brain Trauma Foundation guidelines for preventing secondary injury in critically ill TBI patients.

Similarly, Shehab et al. [7] reached comparable conclusions, noting a need for more educational materials and protocols for managing TBI patients. After implementing an educational program, significant improvements in intensive care unit nurses’ knowledge levels were observed, reducing complication risks among critically ill patients.

### 3.5. Family Support

Reducing agitation levels in critically ill patients is paramount for improving their prognosis. However, conventional methods like medication and physical restraint often come with undesirable side effects and may even exacerbate agitation [29]. According to Sedghi et al. [29], sensory stimulation emerges as an effective and accessible approach to mitigate agitation in patients with reduced consciousness levels. Notably, tactile and auditory stimulation administered by family members proves more efficacious than when performed by nurses. Therefore, families should be regarded not merely as visitors but as integral components of critically ill patients’ recovery journey, encouraged to engage in the stimulation process actively. Promoting family involvement not only aids in reducing patient agitation but also alleviates family members’ anxiety levels [29].

## 4. Discussion

During the scoping review process, fifteen documents were identified that addressed the defined research question, highlighting a scarcity of research in the field. TBI carries a significant global prevalence, accompanied by high mortality and morbidity rates, which contribute to heightened healthcare expenditure [6,7,13,17,22,23,26,29,30].

The selected documents originate from diverse countries across four continents, underscoring a global concern.

Multimodal neuromonitoring is a critical tool in identifying subtle neurophysiological changes that can lead to secondary injury in TBI cases. By applying various monitoring techniques, such as continuous electroencephalography, cerebral perfusion monitoring, and intracranial pressure monitoring, healthcare professionals can closely monitor the neurological status of TBI patients and intervene promptly when necessary [31,32]. It is essential to ensure that healthcare professionals, including nurses, are proficient in utilizing these monitoring modalities and interpreting their findings accurately and swiftly. As frontline caregivers, nurses often play a central role in continuously monitoring TBI patients. Seiler et al. [24] underscore the significance of training nursing staff in continuous EEG monitoring. Such training programs enhance nurses’ knowledge and skills and promote a standardized approach to neuromonitoring in critically ill TBI patients.

Effective control of intracranial pressure is a crucial aspect of managing TBI patients. Elevated intracranial pressure can lead to further neurological damage and worsen patient outcomes [33]. Therefore, healthcare professionals must implement various interventions to regulate intracranial pressure, including pharmacological approaches [12,17,23,26,27], optimizing ventilation [34,35], and positioning strategies [36,37]. Mohammed et al. [30] advocate for normothermia as the recommended body temperature range for critically ill patients with TBI in the context of physiological alterations. However, Tran [26] and Cecil et al. [21] endorse hypothermia as a neuroprotective measure. Conversely, Thompson [25] links hypothermia to an increased risk of mortality. Despite the importance of body temperature fluctuations, nurses may not consistently prioritize it as a critical variable in preventing secondary injury [22].

Nurses play a crucial role in preparing, administering, and monitoring therapeutic interventions for TBI patients. Changes in intravascular volume can significantly impact the condition of critically ill TBI patients, exacerbating their condition [12]. As various authors advocate [12,23,27], maintaining normovolemia is essential for optimal patient outcomes. Authors recommend using crystalloid fluids while cautioning against hypotonic saline solutions [12,23,27]. Additionally, Tran [26] highlights the administration of hypertonic fluids as a neuroprotective measure due to their role in maintaining adequate blood pressure levels in patients with TBI.

Nurses are responsible for ensuring the accurate preparation and administration of fluids and medications. They closely monitor the patient’s response to treatment, including changes in intravascular volume and blood pressure levels [38,39]. Through vigilant observation and regular assessments, nurses can detect early signs of fluid imbalance or adverse reactions to medication, enabling timely intervention to prevent further complications [38,40].

Furthermore, nurses collaborate with other healthcare professionals to develop and implement individualized care plans for TBI patients, considering their unique physiological and neurological needs. They are vital in coordinating care, advocating for patients, and providing education and support to patients and their families throughout treatment [41,42].

Analytical surveillance is the cornerstone of nursing interventions, enabling real-time monitoring and immediate response to patient condition changes. This proactive approach allows nurses to detect subtle variations in physiological parameters and promptly initiate appropriate interventions, often in collaboration with other healthcare professionals [21].

One of the most innovative techniques in analytical surveillance is microdialysis technology, which continuously monitors metabolic markers such as glucose, pyruvate, lactate, and glycerol in the brain tissue [21]. Cecil et al. [21] underscored the significance of this technology in maintaining cellular homeostasis and optimizing brain function. By providing insights into cerebral metabolism, microdialysis allows for early detection of metabolic derangements, facilitating timely interventions to prevent secondary brain injury.

Even with its potential benefits, microdialysis technology also has limitations and drawbacks. One challenge is the procedure’s invasive nature, which requires inserting a microdialysis catheter into the brain tissue [21], posing a risk of infection and tissue damage. Additionally, interpreting microdialysis data requires specialized training and expertise [43,44]. Despite these challenges, integrating microdialysis technology into clinical practice holds promise for improving the management of TBI patients. With ongoing technological advancements and the development of novel monitoring techniques, nurses can leverage these tools to enhance patient care and outcomes.

There is a consensus on the importance of avoiding hyperglycemia and hypoglycemia, although the exact target values remain debated. Cecil et al. [21] suggests maintaining levels between 100 and 180 mg/dL, whereas Boling and Groves [12] propose a broader range of 80–200 mg/dL to mitigate the risk of hypoglycemia. However, Prisco et al. [45] found that early hyperglycemia serves as a significant predictor of mortality and correlates with other factors contributing to secondary injury. Their findings indicate that early hyperglycemia (with values ranging between 126 and 182 mg/dL) indicates an inflammatory response, leading to early cardiovascular and respiratory complications.

Among electrolyte disorders, dysnatremia is emphasized [6,12], with discussions on the various syndromes that can lead to it. Both authors stress the significance of recognizing these differences to apply the most appropriate treatment.

Numerous articles underscore the importance of training nursing staff, significantly enhancing patient outcomes.

Despite progress, there remains a lack of knowledge on the subject, highlighting the pivotal role of training in overcoming this deficiency and reducing secondary injuries [7,24].

All studies encompassed in this scoping review concerning the training of nursing staff in managing trauma victims illustrate the value professionals place on it. Shehab et al. [7] highlight its role in mitigating the risk of severe complications through systematic training. Multiple authors advocate for developing protocols grounded in the latest scientific evidence, significantly impacting patient outcomes [22,30]. Additionally, it is crucial to recognize the importance of task shifting and task sharing at the junior doctor level and among nursing staff. This is particularly critical in services where more senior nurses, such as Advanced Nurse Practitioners, Nurse Specialists, and Consultant Nurses, can effectively coordinate the activities of the staff they manage [46].

Given their complexity, critically ill patients who are victims of TBI require a multifactorial and interdisciplinary approach [30]. This underscores the need for standardization and evidence-based practice to guide the treatment of the patient victim of TBI, thus minimizing the possibility of care omissions and intervention duplications [30].

Family support for TBI victims is pivotal to their recovery. Tactile and auditory stimulation play a fundamental role in reducing agitation among critically ill patients with diminished consciousness levels; as Sedghi et al. [29] highlight, it proves more effective when conducted by family members than professionals.

### Limitations

The limitations of the scoping review should be considered when interpreting its findings. First, there is a risk of publication bias, given that only studies published in English and Portuguese were included. This may have excluded relevant research published in other languages, introducing bias and potentially overlooking essential findings. Second, the fact that we limited our search to six databases and applied a temporal limit may have excluded potentially relevant studies. This could have restricted the scope of the review and overlooked recent developments or studies in databases not included in our search strategy. Additionally, this scoping review did not explore the role of nurses-led service improvement projects for the long-term follow-up of TBI. Although those projects are less prone to prevent secondary brain injury cascade, they may still support patients and their families, ultimately improving the clinical outcome after TBI. Such limitations could impact the comprehensiveness and currency of the findings, potentially affecting the conclusions drawn from the review.

## 5. Conclusions

The research conducted for this scoping review highlights a global concern regarding the prevention of secondary injury in critically ill patients with TBI. However, articles specifically detailing nursing interventions in this area are scarce, indicating a need for further clarification among care providers. There remains ample room for additional research, mainly focusing on nursing interventions’ significant impact on patient outcomes.

Upon reviewing the selected documentation, several critical nursing interventions emerged, including recognizing changes in intracranial pressure, appropriate utilization of multimodal neuromonitoring, and prompt action to address these changes, guided by the principle of “Time is Brain”. Given the multidisciplinary nature of this work, developing action protocols and guidelines is crucial to ensure coordinated treatment efforts, avoiding duplication of interventions and omission of critical aspects of care.

In conclusion, the systematization of care through comprehensive training for healthcare professionals, particularly specialist nurses, is imperative. By promoting training and awareness-raising initiatives, specialist nurses can optimize their interventions and underscore the importance of their role in preventing secondary injury.

Through evidence-based practice, specialized nursing has the potential to significantly reduce mortality and morbidity rates, yielding tangible benefits for both individuals and society.

## Figures and Tables

**Figure 1 jcm-13-02396-f001:**
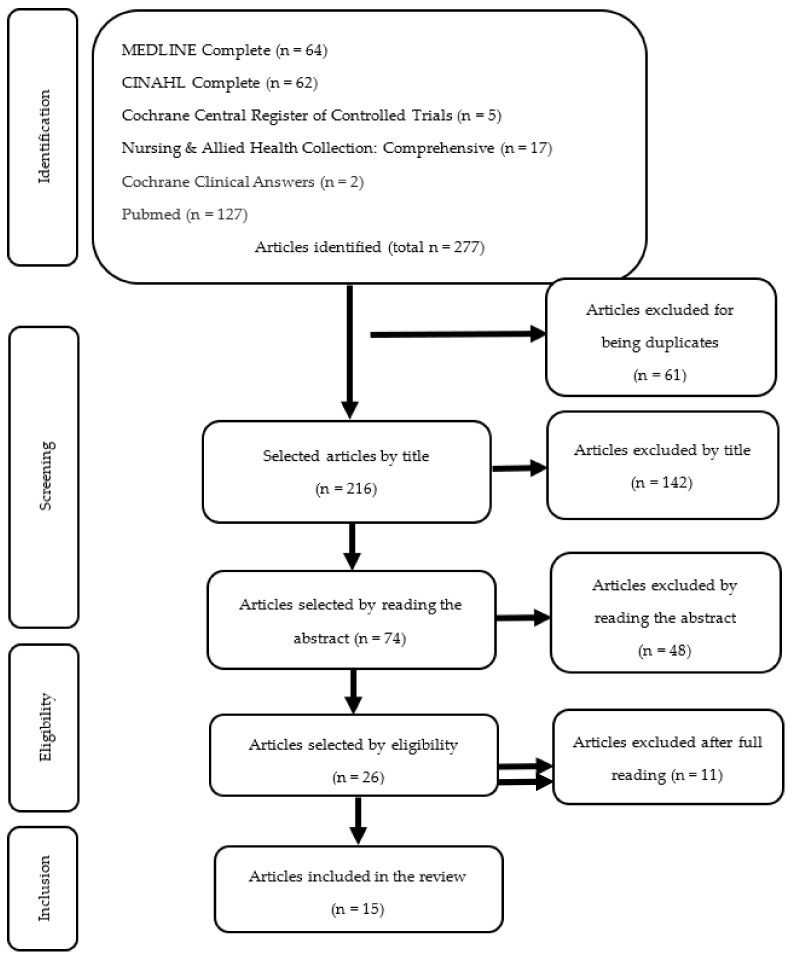
PRISMA flow chart for study selection.

**Table 1 jcm-13-02396-t001:** Inclusion/exclusion criteria.

Parameter	Inclusion Criteria
Participants	Nurses providing care to critically ill patients victims of TBI.
Concept	Studies that explore nursing interventions to prevent secondary injury in TBI.
Context	Studies conducted in an Intensive Care Unit.
Type of studies	Quantitative, qualitative, mixed methods and literature reviews.

**Table 2 jcm-13-02396-t002:** Data extraction and synthesis.

Author/Year/Title/Country	Aim	Study Design	Interventions/Categories
Boling & Groves [12], 2019 Management of Subarachnoid Haemorrhage USA	Discuss the anatomy, epidemiology, and pathophysiology of Subarachnoid hemorrhage: current scientific evidence and protocols in the prevention of secondary brain injury	Literature review	**Neuromonitoring** - Routine use of transcranial Doppler to detect vasospasm - Hyperthermia control - Use of cooling blankets **Analytical surveillance** - Monitoring for hyponatremia **Therapeutics** - Administration of oral Nimodipine - Regular monitoring of the volume of fluid therapy - Use of isotonic crystalloids
Cecil et al. [21], 2011 TBI Advanced Multimodal Neuromonitoring From Theory to Clinical Practice USA	To evaluate the benefits of using multimodal neuromonitoring in a level 1 trauma Intensive Care Unit in the prevention of secondary injury, with emphasis on cerebral microdialysis and monitoring of cerebral blood flow	Literature review	**Neuromonitoring** - Intracranial pressure monitoring and Cerebral Perfusion Pressure calculation - Monitoring cerebral blood flow - Monitoring brain tissue oxygenation: use of *Licox* PBtO2 - Monitoring brain temperature and hypothermia **Analytical surveillance** - Evaluation of metabolic markers: glucose, pyruvate, lactate, and glycerol, with cerebral microdialysis
Iavagnilio, [13], 2011 TBI: Improving the Patient’s Outcome Demands Timely and Accurate Diagnosis USA	Identify warning signs and symptoms of deterioration in Critically Ill Patients victims of TBI and discuss the importance of implementing action protocols	Case study	**Neuromonitoring** - 15/15 min neurological observation: Glasgow Coma Scale, pupillary assessment, respiratory rate, heart rate, blood pressure, and temperature
John & Day [6], 2012 Central Neurogenic Diabetes Insipidus, Syndrome of Inappropriate Secretion of Antidiuretic Hormone and Cerebral Salt-Wasting Syndrome in TBI USA	Identify the causes of Diabetes Insipidus, Syndrome of inappropriate secretion of antidiuretic hormones, and cerebral salt-wasting syndrome and how to control them in TBI	Literature review	**Analytical surveillance** - Monitoring of analytical values that enable the appropriate treatment of each different syndrome
Mcnett et al. [22], 2010 Judgments of Critical Care Nurses About Risk for Secondary Brain Injury USA	Investigate which variable vital signs and situations influence the nursing staff’s perception of secondary injury prevention.	Qualitative study	**Neuromonitoring** - Monitor the variation in oximetry, intracranial pressure and cerebral perfusion pressure values - Implement action protocols to ensure standardized care
Mohamed et al. [30], 2018 The effectiveness of clinical pathway directed care on hospitalization-related outcomes in patients with severe TBI: a quasi-experimental study Egypt	Evaluate the effectiveness of implementing a Clinical pathway in TBI	Quasi-experimental study	**Neuromonitoring** - Monitor body temperature **Analytical surveillance** - Monitor analytical values such as blood glucose and signs of infection **Family Support** - Include the family in the care process
Nyholm et al. [17], 2017 Predictive Factors That May Contribute to Secondary Insults With Nursing Interventions in Adults With TBI USA	Identify the risk/benefit of nursing interventions that prevent or place Critically Ill Patients at risk of secondary injury.	Quantitative study	**Therapeutic** - Administer pre-intervention sedation bolus **Neuromonitoring** - Ensure multimodal monitoring, including intracranial pressure and cerebral perfusion pressure monitoring - Position patient to optimize venous return
Oh et al. [28], 2019 Temporal Patterns and Influential Factors of Blood Glucose Levels During the First 10-day Critical Period After Brain Injury the Republic of Korea	Identify the effects of hyperglycemia in victims of TBI	Quantitative study	**Analytical surveillance** - Document the glycemia pattern - Identify the stress factors that induce hyperglycemia
Schimpf [23], 2012 Diagnosing Increased Intracranial Pressure USA	Develop an algorithm to diagnose the causes of raised intracranial pressure and serve as a guideline for the management	Guideline	**Therapeutic** - Maintain adequate sedation levels **Neuromonitoring** - Keep the headboard elevated at 30–45 degrees - Skin assessment: temperature, infection focus; pupils; ears and nose: blood or Cerebrospinal fluid losses; presence of masses - Glasgow Coma Scale assessment - Monitoring seizures
Sedghi et al. [29], 2020 The Effect of Auditory and Tactile Stimulation by a Family Member on the Level of Agitation in Patients with TBI and Decreased Consciousness: A Quasi-Experimental Study Iran	Determine the effects of sensory stimulation by the family on reducing agitation in PSC with TBI	Quasi-experimental study	**Family support** - Promote the family presence - Encourage the family to talk and touch the patient
Seiler et al. [24], 2012 The Effectiveness of a Staff Education Program on the Use of Continuous EEG With Patients in Neuroscience Intensive Care Units USA	Evaluate the effectiveness of a nursing training program on continuous EEG monitoring in Critically Ill Patients in the Neurocritical Care Intensive Care Unit.	Case study	**Training professionals** - Training the nursing team in monitoring EEG
Shehab et al. [7], 2018 Impact of an Educational Program on Nurses’ Knowledge and Practice Regarding Care of TBI Patients at Intensive Care Unit at Suez Canal University Hospital Egypt	Evaluate the impact of an educational program in the treatment of Critically Ill Patients who are victims of TBI	Quasi-experimental study	**Training professionals** - Training for Intensive Care Unit nurses in order to prevent the development of secondary injury
Thompson et al. [25], 2010 Hypothermia and Rapid Rewarming Is Associated with Worse Outcome Following TBI USA	Determine the prevalence of hypothermia upon admission to the emergency department, the effect of hypothermia, and the speed of rewarming on the outcome of Critically Ill Patients victims of TBI	Case study	**Neuromonitoring** - Assess body temperature upon admission to the Emergency Department - Maintain vigilance until reaching 36.5
Tran [26], 2014 Understanding the Pathophysiology of TBI and the Mechanisms of Action of Neuroprotective Interventions USA	Understand the pathophysiology of TBI and identify interventions to prevent secondary injury	Literature review	**Therapeutic** - Use of hyperosmolarity (mannitol and hypertonic saline) - Administration of Statins - Administration of cyclosporine A
van der Jagt [27], 2016 Fluid management of the neurological patient: a concise review Netherlands	Identify the amount and type of fluid therapy best suited to prevent secondary injury	Literature review	**Therapeutic** - Fluid monitoring: indications and contraindications **Neuromonitoring** - Hypothermia screening - Hemodynamic monitoring

## Data Availability

The data presented in this study are available on request from the first author.

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
