# Peer review of "Nursing Interventions to Prevent Secondary Injury in Critically Ill Patients with Traumatic Brain Injury: A Scoping Review"

_jcm, 2024, doi:10.3390/jcm13082396_

Round 1
Reviewer 1 Report
Comments and Suggestions for Authors
Authors present a scoping review on nursing interventions to prevent secondary injury in critically ill patients with traumatic brain injury. A total of 15 manuscripts underwent final analysis, and nursing aspects were analyzed in neuromonitoring, therapeutic interventions, analytical surveillance methods, professional training initiatives, and family support measures. It is unclear how all studies in Table 1. really coincide with specific aspects of nursing - providing care as in prompt recognition of neurological decline is a vital part of care for TBI patients, but this is true for most of acute neurosurgical conditions. Studies like Shehab et al, Seiler et al, Mohamed et al deal specific with the subject, but others concentrate more on interventional aspects in TBI. Authors should emphasize more the specific nursing aspects, as it was done in Section 3.4. Include specific training for ICU nurses in dealing with TBI.
Comments on the Quality of English LanguageModerate changes.
Author Response
We want to thank the reviewer for the thoughtful comments and efforts toward improving and building on the manuscript. We have incorporated changes to reflect all the suggestions provided. We have highlighted these changes within the manuscript.
Reviewer 1
Comment:
Authors present a scoping review on nursing interventions to prevent secondary injury in critically ill patients with traumatic brain injury. A total of 15 manuscripts underwent final analysis, and nursing aspects were analyzed in neuromonitoring, therapeutic interventions, analytical surveillance methods, professional training initiatives, and family support measures. It is unclear how all studies in Table 1. really coincide with specific aspects of nursing - providing care as in prompt recognition of neurological decline is a vital part of care for TBI patients, but this is true for most of acute neurosurgical conditions. Studies like Shehab et al, Seiler et al, Mohamed et al deal specific with the subject, but others concentrate more on interventional aspects in TBI. Authors should emphasize more the specific nursing aspects, as it was done in Section 3.4. Include specific training for ICU nurses in dealing with TBI.
Answer to Reviewer
Thank you for your insightful feedback. We have enhanced the presentation of our results to underscore the pivotal role of nurses, especially in neuromonitoring, therapeutics, and analytical surveillance. In the discussion section, we have emphasized nursing interventions, highlighting their specific contributions and nuances.
Reviewer 2 Report
Comments and Suggestions for Authors
The authors conducted an interesting scoping review meant to explore the evidence regarding nursing interventions in preventing secondary injury cascade in critically ill patients presenting with traumatic brain injury.
I commend the authors for their research question with covers a very wide spectrum including pre-hospital care, Emergency Departments and Neuro-Intensive Care Unit settings. To provide a roadmap to further improve their already very good work, I have the following comments and requests for the authors:
1) From a methodological perspective, I appreciate the comments about adhering to the PRISMA ScR checklist, however I'm not sure that there is a value in saying that their review was not registered. In fact, while the PROSPERO registry is universally accepted for registration of systematic reviews, it does not accept scoping reviews, and the current registration process in other platforms (i.e. OSF) for scoping review is not as robust and widely accepted as it happens to be for systematic ones. I would therefore modify the sentence in line 74.
2) Using the PCC framework is very appropriate for qualitative studies, which represent the majority of research designed and conducted by nurses. I would highlight this aspect to support the choice of using PCC rather than PICO, accordingly I would modify the sentence in line 99.
3) In section 3.1. I would suggest adding a line on Near Infrared Spectroscopy as a valuable tool in patients with TBI To this regard, there is a very nice study published this year that demonstrates how nursing postoperative neurosurgical patients with head of bed elevation beyond 30° might be desired at times to prevent pulmonary complications. The authors used a mix of neuromonitoring techniques including NIRS to make their point. I would suggest adding this article to support your paragraph on NIRS: Baskar N, Sethuraman M, Praveen R, Hrishi AP, Vimala S, Prathapadas U, Abraham M. Evaluation of Cerebral Perfusion Pressure, Cerebral Blood Flow, and Cerebral Oxygenation at Different Head of Bed Positions Using Transcranial Doppler and Near-Infrared Spectroscopy in Postoperative Neurosurgical Patients. Cureus. 2024;16(1):e51923. doi: 10.7759/cureus.51923.
4) In section 3.3. you correctly pointed out that hyperglycemia can be deleterious for TBI patients (lines 270-275). I would suggest adding the following reference from Prisco et al. to reinforce your point and provide values supporting the view that early hyperglycemia (with values above 147 mg/DL) is a major predictor of mortality and correlates with other factors responsible for secondary injury. Early hyperglycemia (values between 126 and 182 mg/dL) seems to be a marker of inflammatory reaction responsible for early cardiovascular and respiratory impairment: Prisco L, Iscra F, Ganau M, Berlot G. Early predictive factors on mortality in head injured patients: a retrospective analysis of 112 traumatic brain injured patients. J Neurosurg Sci. 201;56(2):131-6.
5) In the discussion I would add a sentence about the importance of task shifting and task sharing not only at junior doctors level but also at nursing staff level. This is critically important in services where more senior nurses (Advanced Nurse Practitioners, Nurse Specialists and Consultant Nurses) can coordinate activities of the staff they managed. I would use the following reference to make this point: Robertson FC, Esene IN, Kolias AG, Khan T, Rosseau G, Gormley WB, Park KB, Broekman MLD; Global Neurosurgery Survey Collaborators. Global Perspectives on Task Shifting and Task Sharing in Neurosurgery. World Neurosurg X. 2019;6:100060. doi: 10.1016/j.wnsx.2019.100060.
6) In section 4.1. I would also add that this scoping review did not explore the role of nurses-led service improvement projects for the long term follow up of TBI. Although those projects are less prone to prevent secondary brain injury cascade, they may still provide incredible support to patients and their families, ultimately improving the clinical outcome after TBI.
I hope that the authors could find the 6 points above useful for their revision and I look forward to receive their updated manuscript.
Author Response
Reviewer 2
We want to thank the reviewer for the thoughtful comments and efforts toward improving and building on the manuscript. We have incorporated changes to reflect all the suggestions provided. We have highlighted these changes within the manuscript.
Comment:
The authors conducted an interesting scoping review meant to explore the evidence regarding nursing interventions in preventing secondary injury cascade in critically ill patients presenting with traumatic brain injury.
I commend the authors for their research question with covers a very wide spectrum including pre-hospital care, Emergency Departments and Neuro-Intensive Care Unit settings. To provide a roadmap to further improve their already very good work, I have the following comments and requests for the authors:
1) From a methodological perspective, I appreciate the comments about adhering to the PRISMA ScR checklist, however I'm not sure that there is a value in saying that their review was not registered. In fact, while the PROSPERO registry is universally accepted for registration of systematic reviews, it does not accept scoping reviews, and the current registration process in other platforms (i.e. OSF) for scoping review is not as robust and widely accepted as it happens to be for systematic ones. I would therefore modify the sentence in line 74.
Answer to Reviewer
We have revised the sentence and removed the information.
Comment:
2) Using the PCC framework is very appropriate for qualitative studies, which represent the majority of research designed and conducted by nurses. I would highlight this aspect to support the choice of using PCC rather than PICO, accordingly I would modify the sentence in line 99.
Answer to Reviewer
The editor has requested that we shorten the methodology. While I agree with the reviewer, we feel that including this information may not significantly enhance the review.
Comment:
3) In section 3.1. I would suggest adding a line on Near Infrared Spectroscopy as a valuable tool in patients with TBI To this regard, there is a very nice study published this year that demonstrates how nursing postoperative neurosurgical patients with head of bed elevation beyond 30° might be desired at times to prevent pulmonary complications. The authors used a mix of neuromonitoring techniques including NIRS to make their point. I would suggest adding this article to support your paragraph on NIRS: Baskar N, Sethuraman M, Praveen R, Hrishi AP, Vimala S, Prathapadas U, Abraham M. Evaluation of Cerebral Perfusion Pressure, Cerebral Blood Flow, and Cerebral Oxygenation at Different Head of Bed Positions Using Transcranial Doppler and Near-Infrared Spectroscopy in Postoperative Neurosurgical Patients. Cureus. 2024;16(1):e51923. doi: 10.7759/cureus.51923.
4) In section 3.3. you correctly pointed out that hyperglycemia can be deleterious for TBI patients (lines 270-275). I would suggest adding the following reference from Prisco et al. to reinforce your point and provide values supporting the view that early hyperglycemia (with values above 147 mg/DL) is a major predictor of mortality and correlates with other factors responsible for secondary injury. Early hyperglycemia (values between 126 and 182 mg/dL) seems to be a marker of inflammatory reaction responsible for early cardiovascular and respiratory impairment: Prisco L, Iscra F, Ganau M, Berlot G. Early predictive factors on mortality in head injured patients: a retrospective analysis of 112 traumatic brain injured patients. J Neurosurg Sci. 201;56(2):131-6.
Answer to Reviewer
We thank the reviewer for the suggestion. However, in the results section of the scoping review, we can only present data from studies that have undergone the selection process. While the information provided is relevant, we cannot include such data.
Comment:
5) In the discussion I would add a sentence about the importance of task shifting and task sharing not only at junior doctors level but also at nursing staff level. This is critically important in services where more senior nurses (Advanced Nurse Practitioners, Nurse Specialists and Consultant Nurses) can coordinate activities of the staff they managed. I would use the following reference to make this point: Robertson FC, Esene IN, Kolias AG, Khan T, Rosseau G, Gormley WB, Park KB, Broekman MLD; Global Neurosurgery Survey Collaborators. Global Perspectives on Task Shifting and Task Sharing in Neurosurgery. World Neurosurg X. 2019;6:100060. doi: 10.1016/j.wnsx.2019.100060.
Answer to Reviewer
We have introduced a paragraph in the discussion focusing on the importance of task shifting and sharing.
Comment:
6) In section 4.1. I would also add that this scoping review did not explore the role of nurses-led service improvement projects for the long term follow up of TBI. Although those projects are less prone to prevent secondary brain injury cascade, they may still provide incredible support to patients and their families, ultimately improving the clinical outcome after TBI.
I hope that the authors could find the 6 points above useful for their revision and I look forward to receive their updated manuscript.
Answer to Reviewer
We have added the information you provided to section 4.1. Thank you for your valuable input.
Reviewer 3 Report
Comments and Suggestions for Authors
Dear Authors,
I have carefully read your paper, which aim was to map and analyze the existing scientific evidence on nursing interventions aimed at preventing secondary injuries in critically ill patients with traumatic brain injury.
This study is systematic review and it has used structured approach according to PRISMA guidelines. Flow chart is done well showing process of applying inclusion/exclusion criteria. The findings have presented nursing interventions in the investigated field. The article is well written with results presented clearly and easy to understand. However, scientific significance may not be high. Improvements in the Discussion are needed. You have presented findings of other authors, but still, you have not given your comments on whether these strategies may be of clinical interest or not, possible or not in the daily praxis…what can be problematic…etc
I suggest revision.
Comments on the Quality of English LanguageMinor corrections are needed.
Author Response
Reviewer 3
We want to thank the reviewer for the thoughtful comments and efforts toward improving and building on the manuscript. We have incorporated changes to reflect all the suggestions provided. We have highlighted these changes within the manuscript.
Comment:
I have carefully read your paper, which aim was to map and analyze the existing scientific evidence on nursing interventions aimed at preventing secondary injuries in critically ill patients with traumatic brain injury.
This study is systematic review and it has used structured approach according to PRISMA guidelines. Flow chart is done well showing process of applying inclusion/exclusion criteria. The findings have presented nursing interventions in the investigated field. The article is well written with results presented clearly and easy to understand. However, scientific significance may not be high. Improvements in the Discussion are needed. You have presented findings of other authors, but still, you have not given your comments on whether these strategies may be of clinical interest or not, possible or not in the daily praxis…what can be problematic…etc
I suggest revision.
Answer to Reviewer
Thank you for your insightful feedback. We have enhanced the presentation of our results to underscore the pivotal role of nurses, especially in neuromonitoring, therapeutics, and analytical surveillance. In the discussion section, we have emphasized nursing interventions, highlighting their specific contributions and nuances.
Round 2
Reviewer 1 Report
Comments and Suggestions for Authors
Authors have sufficiently responded to reviewer remarks.
Author Response
Thank you
Reviewer 2 Report
Comments and Suggestions for Authors
Unfortunately the paragraph on hyper-hypoglycemia has not been improved. By discussing only articles with nursing involvemente the authors risk to provide misleading statements and downgrade the usefulness of this study. While I see the poinylt of including in the table only studies that were shortlisted during your screening procress you should expand the discussion by using appropriate referencing. At least the study from Prisco et al. Should therefore be mentioned.
Author Response
Reviewer 2
We want to thank the reviewer for the thoughtful comments and efforts toward improving and building on the manuscript. We have incorporated changes to reflect all the suggestions provided. We have highlighted these changes within the manuscript.
Comment:
Thank you for the feedback and suggestions. We have made changes to the discussion to reflect the reviewer's recommendation and added a reference to the study by Prisco et al. (2012) to ensure a more comprehensive and accurate analysis of the topic. Once again, we appreciate your contribution to improving the quality of our work.